# Value Residual Learning For Alleviating Attention Concentration In Transformers

## Abstract

Transformers can capture long-range dependencies using self-attention, allowing tokens to attend to all others directly. However, stacking multiple attention layers leads to attention concentration. One natural way to address this issue is to use cross-layer attention, allowing information from earlier layers to be directly accessible to later layers. However, this approach is computationally expensive. To address this problem, we propose Transformer with residual value (ResFormer) which approximates cross-layer attention through adding a residual connection from the values of the the first layer to all subsequent layers. Based on this method, one variant is the Transformer with single layer value (SVFormer), where all layers share the same value embedding from first layer, reducing the $KV$ cache by nearly 50%. Comprehensive empirical evidence demonstrates that ResFormer mitigates attention concentration problem in deeper layers and enhances representation across most layers, outperforming the vanilla Transformer, DenseFormer, and NeuTRENO in training error as well as downstream tasks. SVFormer trains significantly faster than the vanilla Transformer and performs better than other methods like GQA and CLA, with performance influenced by sequence length and cumulative learning rate.

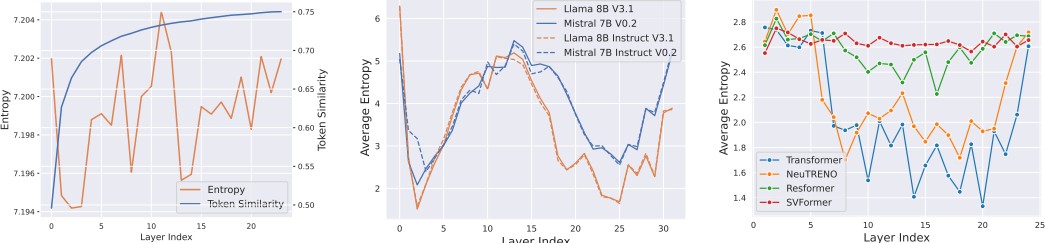

Figure 1: (Left) The average entropy of token importance and the average hidden-state similarity for a randomly initialized 468M model. (Middle) The average entropy of token importance across layers in Llama (8B) (Dubey et al., 2024) and Mistral (7B) (Jiang et al., 2023). (Right) The average entropy of token importance across layers in ResFormer *vs.* the vanilla Transformer, where token importance is derived from the attention matrix. Lower entropy indicates more focused attention on specific tokens. More details can be found in Eqn. 11.

## 1 Introduction

The Transformer (Vaswani et al., 2017) model has become one of the leading architectures in recent years, excelling in both language modeling (Devlin et al., 2019; Lan et al., 2020; Brown et al., 2020) and computer vision tasks (Dosovitskiy et al., 2021). The discovery of scaling laws (Hoffmann et al., 2022; Kaplan et al., 2020) has driven the pursuit of larger Transformer models by increasing network depth and width. Training large models presents significant challenges. Balancing the depth and width of a Transformer model within a fixed parameter budget is particularly difficult. While research indicates that deeper models generalize more compositionally than shallower ones (Petty et al., 2024), the training and deployment of deep models remain problematic. Although Transformers use residual connections (He et al., 2016) to address the vanishing gradient issue, training very

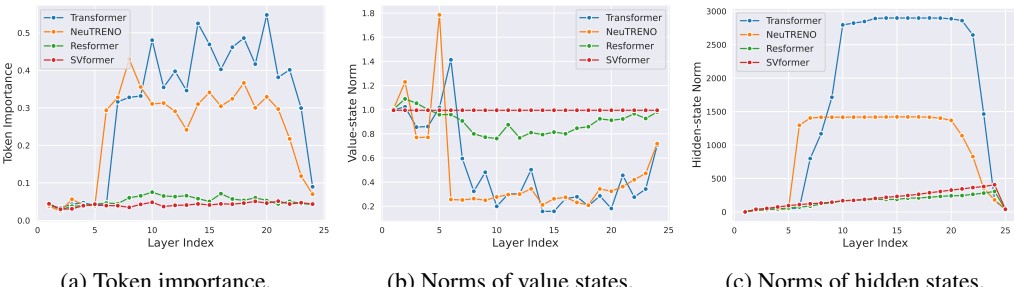

(a) Token importance.  (b) Norms of value states.  (c) Norms of hidden states.

Figure 2: The token importance (Xiao et al., 2024), value-state norms (Guo et al., 2024b), and hidden-state norms (Sun et al., 2024) of the first token across layers of 468M models. More Visualization results are available in Appendix A.2.

deep Transformers is still challenging. For example, a 32-layer Vision Transformer (ViT) may perform worse than a 24-layer one (Zhou et al., 2021). This is mainly due to the smoothing mechanism of attention (Shi et al., 2022), which can lead to an over-smoothing effect (Nguyen et al., 2023) where the token representations become the same as the model's depth increases.

Existing solutions to alleviate the over-smoothing problem in Transformer include adding extra regularizers (Nguyen et al., 2023; Shi et al., 2022) and optimizing the information flow within the model (Pagliardini et al., 2024). During the era of convolutional neural network architectures, Stochastic Depth (Huang et al., 2016) reduces the likelihood of over-smoothing by randomly dropping layers during training and DenseNet (Huang et al., 2017) mitigates the impact of over-smoothing by allowing each layer to directly access the hidden states of all preceding layers. Recently, DenseFormer (Pagliardini et al., 2024) adopts the idea of DenseNet when training Transformer. Additionally, NeuTRENO (Nguyen et al., 2023) alleviates over-smoothing through incorporating the difference between the value vectors of the first layer and the current layer to the attention output.

In this paper, we address the problem of multi-layer attention from another perspective. We introduce the phenomenon of attention concentration, which describes how a model's attention increasingly focuses on fewer tokens. We quantify the degree of attention concentration using the entropy of the distribution of token importance, where lower entropy indicates a more pronounced concentration. Unlike over-smoothing, which is inherent to model architecture, attention concentration emerges during training. Fig. 1 (Left) shows that randomly initialized models exhibit over-smoothing but not attention concentration. Trained ViT models often focus on low-informative background areas (Darcet et al., 2024), while language models concentrate on low-semantic tokens (Sun et al., 2024), particularly the start token (attention sink (Xiao et al., 2024)). While previous studies analyzed single-layer attention patterns, our research reveals a "concentration - dispersion - concentration" pattern in deep models, as shown in Fig. 1 (Middle), suggesting potential loss of information during concentrated phases. The analysis of over-smoothing is available in Appendix A.1.

Mitigating attention concentration can lead to more interpretable attention maps and potentially improve downstream task performance (Darcet et al., 2024). This phenomenon typically emerges after the second or third network layer and is associated with value-state drains (decreased magnitude of value states) (Guo et al., 2024b), and hidden-state peaks (increased magnitude of hidden states) (Sun et al., 2024). Guo et al. (2024a) shows a mutual reinforcement mechanism exists between value-state drains and attention concentration. Recent studies have linked this to implicit biases during pretraining, with most existing solutions focusing on the use of additional tokens (registers) (Darcet et al., 2024) or additional keys and values (explicit attention bias) (Sun et al., 2024) to redirect this.

Given that the first layer always shows no attention concentration, an effective method is to use cross-layer attention on information from this layer. However, due to computational costs, we propose ResFormer as an efficient alternative. ResFormer applies a residual connection between the value vectors of the current layer and the first layer before the attention operation. Unlike cross-layer attention, ResFormer indirectly mitigates attention concentration. It leverages the absence of value-state drains in the first layer by introducing a value residual connection. This alleviates value-state drains in deeper layers, thereby disrupting the mutual reinforcement between attention concentration and value-state drains, as shown in Fig. 1 (Right) and Fig. 2.

During inference, deep networks require substantial $KV$ cache, severely impacting model deployment (Xiao et al., 2024). Existing $KV$-efficient methods often process keys and values simultane-

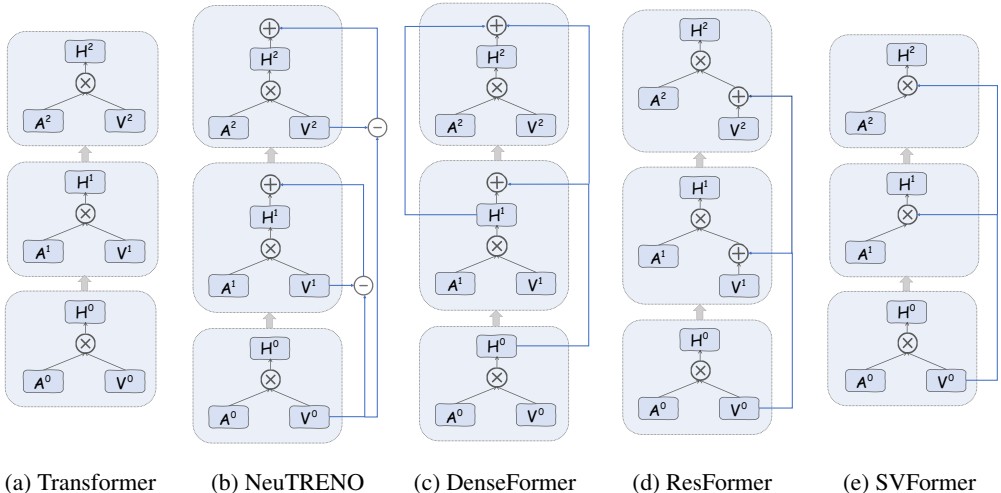

(a) Transformer     (b) NeuTRENO     (c) DenseFormer     (d) ResFormer     (e) SVFormer

Figure 3: Simplified illustration of the vanilla Transformer, NeuTRENO, DenseFormer, ResFormer, and SVFormer, with only three-layer structures and no operations other than attention. $A^i$, $V^i$, and $H^i$ denote the attention matrix, value vectors, and attention outputs at the $i$-th layer, respectively. $\oplus$, $\ominus$, and $\otimes$ represent standard matrix addition, subtraction, and multiplication, respectively.

ously. Building on ResFormer, we decouple the value from the attention operation and propose a new kind of Transformer with single layer value (SVFormer). In SVFormer, the queries and keys of all layers share the value from the first layer, and thus it can also alleviate attention concentration.

We experiment on a 20B SlimPajama sub-sampled dataset, using settings similar to popular large language models (Wei et al., 2023; Dubey et al., 2024; Kaplan et al., 2020). We compare different models by their relative training curves against the vanilla Transformer. Results show that Res-Former outperforms the vanilla Transformer, DenseFormer, and NeuTRENO. ResFormer achieves equivalent validation loss with 10.4% fewer model parameters and 13.6% less training data compared to Transformer, while maintaining similar memory usage and computational cost. Besides, SVFormer, while reducing the $KV$-cache by nearly half, requires a 12.2% increase in parameters to achieve the same validation loss as Transformer. And the performance of SVFormer is better when the training sequence length is longer. It further reduces the $KV$ cache when integrated with classical method GQA (Ainslie et al., 2023).

## 2 RELATED WORK

### 2.1 SHORTCUT CONNECTIONS FOR BETTER INFORMATION FLOW

Deep learning models often consist of multiple layers, posing a challenge to minimize information loss during transmission. ResNet (He et al., 2016) mitigates the vanishing gradient problem with identity connections. Stochastic Depth (Huang et al., 2016) enhances training by randomly dropping layers. DenseNet (Huang et al., 2017) allows subsequent layers to directly access the hidden states of all preceding layers. These two methods further enhance the information flow after ResNet.

Related research indicates that for advanced Transformer architectures, although increasing depth continues to yield performance improvements in language modeling tasks, the gains become less significant with further increases (Petty et al., 2024). Furthermore, Zhou et al. (2021) illustrates that a 32-layer ViT underperforms a 24-layer ViT. Depth-Wise Attention (ElNokrashy et al., 2024) allows each query to access the key and value at the same position from previous layers through an attention-like mechanism before the output layer. DenseFormer (Pagliardini et al., 2024) integrates weighted fusion of outputs from all preceding layers after each layer. To explore why increasing depth in Transformers does not yield expected gains, Wang et al. (2022) finds that self-attention acts as a low-pass filter, smoothing token representations in ViTs. Additionally, Shi et al. (2022) investigates over-smoothing from a graph perspective in BERT-based language modeling tasks. Neu-

TRENO (Nguyen et al., 2023) adds the difference between the value vectors of the first and current layers to each layer's attention output and significantly alleviates the over-smoothing problem.

In contrast to these methods, ResFormer accesses and integrates information from previous layers prior to the attention operation, as illustrated in Fig. 3. Moreover, it does not require the selection or tuning of additional hyperparameters.

## 2.2 $KV$ CACHE COMPRESSING

The $KV$ cache is a key factor limiting the efficiency of long-text model inference. Research in this area can be broadly classified into Transformer-based methods, which target redundant information in Transformer models, and non-Transformer methods, which mainly addresses the quadratic time complexity of attention with respect to sequence length.

For non-Transformer methods, Mamba (Gu & Dao, 2023) and RWKV (Peng et al., 2023) are two popular works. They replace the original softmax-based attention with SSM (Gu et al., 2021) and AFT (Zhai et al., 2021) mechanisms, respectively. Besides, several approaches have been proposed to enhance models' ability to process long texts while reducing the reliance on $KV$ cache. Dai et al. (2019) advocates segmenting long texts into smaller parts for attention computation. Furthermore, Munkhdalai et al. (2024) uses a fixed-size memory matrix for storing and retrieving past information.

Transformer-based methods can be categorized into three main groups. The first group consists of post-training methods like SnapKV (Li et al., 2024) and ThinK (Xu et al., 2024), which compress $KV$ cache during inference based on attention matrices at token or hidden dimension levels. The second group focuses on quantization and adopts low-precision $KV$ cache quantization rather than completely eliminating them (Hooper et al., 2024). The third group aims to maximize the efficiency of attention-based models via parameter or activation value sharing. The most representative works include Multi-Query Attention (Shazeer, 2019) and Grouped-Query Attention (Ainslie et al., 2023) which suggest to share key and value across a group of queries. MLKV (Zuhri et al., 2024) further suggest to share keys and values for queries across layers and MLA (Liu et al., 2024) introduces low-rank projection when processing keys and values. Besides, CLA (Brandon et al., 2024) and LISA (Mu et al., 2024) respectively point out that we can reuse keys, values, or the attention matrix across layers to reduce redundancy between layers. While these methods typically process both key and value simultaneously, SVFormer is the first approach to decouple value from query and key during attention computation. Moreover, it is compatible with other methods like GQA.

## 3 METHOD

### 3.1 MOTIVATION: INFORMATION TRANSFER VIA CROSS LAYER ATTENTION

Let $\mathbf{H}_n \in \mathbb{R}^{l \times d}$ be the input hidden state of the $n$-th layer, where $l$ denotes the sequence length and $d$ is the dimension size. In standard attention, the hidden state $\mathbf{H}_n$ will be firstly projected into $\mathbf{Q}_n, \mathbf{K}_n, \mathbf{V}_n \in \mathbb{R}^{l \times d}$ through three linear projections $\mathbf{W^Q}, \mathbf{W^K}, \mathbf{W^V} \in \mathbb{R}^{d \times d}$ respectively. For simplicity, we introduce dot-product attention of layer $n$ as

$$\text{Attention}(\mathbf{Q}_n, \mathbf{K}_n, \mathbf{V}_n) = \text{Softmax}(\frac{\mathbf{Q}_n \mathbf{K}_n^T}{\sqrt{d}})\mathbf{V}_n. \tag{1}$$

An ideal way to incorporate previous layers' information is cross layer attention. The attention mechanism naturally extracts relevant information from previous layers. If these layers contain low-quality information, the similarity between the current layer's query and the previous layers' keys will be low, thus minimizing negative impacts. Given $m < n$ and the information $(\mathbf{Q}_m, \mathbf{K}_m, \mathbf{V}_m)$ of $m$-th layer, the cross layer mechanism calculates the attention output $\mathbf{U}_n$ of $n$-th layer by the following attention formula:

$$\mathbf{U}_n = \text{Softmax}\left(\mathbf{Q}_n \text{Concat}(\mathbf{K}_n, \mathbf{K}_m)^T / \sqrt{d}\right) \text{Concat}(\mathbf{V}_n, \mathbf{V}_m). \tag{2}$$

In practice, cross-layer attention enhances feature fusion by allowing information to flow between layers, capturing both intra-layer and inter-layer dependencies. However, this approach introduces additional computational overhead due to the concatenation of keys and values from multiple layers. For example, in scenarios described by Eqn. 2, the overall computational complexity of the model nearly doubles compared with vanilla attention described in Eqn. 1.

## 3.2 Efficient Cross Layer Attention

To solve this problem, we propose to replace the $\mathbf{K}_m$ with $\mathbf{K}_n$ in Eqn. 2, as shown in Eqn. 3.

$$\mathbf{U}_n \approx \mathrm{Softmax}\left(\mathbf{Q}_n\,\mathrm{Concat}(\mathbf{K}_n, \mathbf{K}_n)^T/\sqrt{d}\right)\mathrm{Concat}(\mathbf{V}_n, \mathbf{V}_m) \quad (3)$$

$$= \frac{1}{2}\mathrm{Softmax}\left(\mathbf{Q}_n\mathbf{K}_n^T/\sqrt{d}\right)(\mathbf{V}_n + \mathbf{V}_m). \quad (4)$$

Utilizing the concept of block matrices, Eqn. 3 can be further simplified into Eqn. 4. This simplification converts the concatenation operation of the two value matrices into an addition operation. Compared to Eqn. 1, this new method only brings a minimal increase in computational complexity while still leveraging the information from the $m$-th layer in the $n$-th layer. Furthermore, Eqn. 4 can be generalized to incorporate cross-layer attention across all preceding $n-1$ layers as follows:

$$\mathbf{U}_n \approx \frac{1}{n}\mathbf{A}_n\sum_{i=1}^{n}\mathbf{V}_i. \quad (5)$$

where $\mathbf{A}_n$ denotes the original attention matrix for layer $n$. From the perspective of information propagation, model described by Eqn. 3 projects the historical values into the current layer's embedding space using the current layer's attention as a weight matrix. For example, a naive approach would be to perform identity mapping, as described by

$$\mathbf{U}_n = \mathbf{A}_n\mathbf{V}_n + \frac{1}{n-1}\sum_{i=1}^{n-1}\mathbf{V}_i. \quad (6)$$

To evaluate the approximation effect of replacing the $\mathbf{K}_m$ with $\mathbf{K}_n$, we randomly select 1,000 pre-training data samples. For each layer of a trained baseline model, assuming cross-layer attention is required for each layer with respect to the previous one, we calculate the cosine similarity between the outputs from Eqn. 2 and Eqn. 4. We also calculate the cosine similarity between the outputs from Eqn. 2 and Eqn. 6 for comparison. Fig. 4 shows that our proposed method provides a good approximation for cross-layer attention.

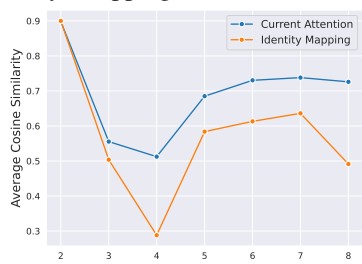

Figure 4: Average token similarity between layer outputs. Lines show similarity of outputs using current attention (Eqn. 4) or identity attention (Eqn. 6) compared to the one using cross-layer attention (Eqn. 2).

## 3.3 Transformer with Residual Value

Based on Eqn. 5, we propose a variant of Transformer with residual value (ResFormer) which only chooses first layer as the target of cross layer attention since the first layer contains all basic information of each token. The analysis of entropy in Fig. 1 (Right) supports this point, indicating that attention tends to be relatively dispersed across different tokens in the initial layers of the model. The attention mechanism of ResFormer can be formulated as

$$\mathbf{U}_n = \frac{1}{2}\mathbf{A}_n(\mathbf{V}_n + \mathbf{V}_1). \quad (7)$$

where $n \geq 2$ and standard attention is applied in the first layer. From the training perspective, it explicitly learns a residual mapping instead of directly learning the desired underlying mapping and that's why we call it ResFormer.

## 3.4 A unified View Of NeuTRENO and DenseFormer

Using our framework, the NeuTRENO can be defined as

$$\mathbf{U}_n = \left(\mathbf{A}_n - \boldsymbol{\lambda}\mathbf{I}\right)\mathbf{V}_n + \boldsymbol{\lambda}\mathbf{V}_1. \quad (8)$$

where $\mathbf{I}$ denotes the identity matrix and $\boldsymbol{\lambda}$ is a hyper-parameter. It can be found that the term of $\boldsymbol{\lambda}\mathbf{I}$ may have certain negative impact on the learning of original attention. If we ignore the attention output projection and the MLP layer, DenseFormer can also be modeled within our framework as

$$\mathbf{U}_n = \sum_{i=1}^{n}\boldsymbol{\alpha}_i\mathbf{A}_i\mathbf{V}_i. \quad (9)$$

where $\{\alpha_i\}_{i=1}^n$ is a set of hyper-parameters. DenseFormer uses attention matrix of previous layer as the weight matrix of projecting values but this is not aligned with the concept shown in Eqn. 3.

### 3.5 SVFORMER: SINGLE-LAYER VALUE FOR HALF $KV$ CACHE

After ResFormer, a natural idea is whether we can remove the value vectors in each layer and have all layers share the value vectors from the first layer. We call this method SVFormer. Similar to ResFormer, SVFormer still adopts standard attention in the first layer and obtain the attention output $\mathbf{U_n}$ for $n$-th layer where $n \geq 2$ through

$$\mathbf{U}_n = \mathbf{A}_n\mathbf{V}_1. \qquad (10)$$

Compared to previous methods, SVFormer is the first method that decouple value vectors from attention. Its main advantage is that it only requires computing and storing the value vectors for the first layer, saving nearly half of the $KV$ cache during inference. Similar methods like CLA reduce $KV$ cache by sharing both of the key and value vectors every two layers. However, the results in Fig. 5 show that sharing values has less negative impact compared with sharing keys.

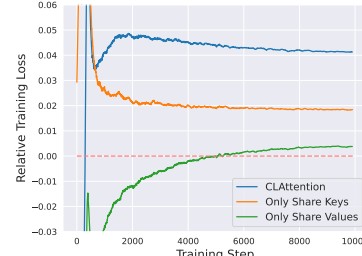

Figure 5: Ablation study of sharing different parts of attention every two layers.

## 4 PRETRAIN EXPERIMENTS

### 4.1 SETTING

#### 4.1.1 TRAINING DETAILS

Following Brandon et al. (2024), we choose the Llama-like architecture and SlimPajama (Soboleva et al., 2023) data for main experiments. Specifically, the architecture includes pre-normalization, SwiGLU activations (Shazeer, 2020), rotary position embedding (Su et al., 2024), and no dropout. For slimpajama, we randomly sample nearly 20B tokens according to the original data distribution of seven domains during training and adopt tokenizer used for "RedPajama-INCITE-7B-Base". The details of training data can be found in Table 2 in Appendix.

Unless otherwise noted, we train all models using AdamW optimizer with 0.1 weight decay, $\beta_1 = 0.9$, $\beta_2 = 0.95$ and the max grad norm 1.0. The batch size is set to be around 2M tokens (Zhang et al., 2024) with a sequence length of 2,048 and the total steps is fixed 10,000 steps (Kaplan et al., 2020). We adopt linear learning rate warmup for the first 1,200 steps with the initial learning rate and the peak learning rate to be 1e-7 and 6e-4 respectively. The cosine decay schedule gradually decays to 10% of the peak learning rate by the end of training (Zhou et al., 2024; Wei et al., 2023). The detailed hyperparameters for models of various sizes and different training sequence lengths in Appendix A.5. Moreover, All models are trained with 8 Nvidia A100 80G GPUs using mixed-precision training in FP16. We adopt deepspeed zero-2 optimizer and flash attention mechanism.

#### 4.1.2 RELATIVE TRAINING LOSS CURVE ON SLIMPAJAMA

We trained all models for only one epoch on SlimPajama subsets, and primarily use training loss to compare different models. Furthermore, we use the relative training loss curve for better visualizing the difference among different models from the perspective of loss landscape. Specifically, for each method, we will subtract the smoothed training curve of the vanilla Transformer, obtained under the same experimental settings, from the smoothed training curves of the method. The smoothing is done using a window size of 10 steps or 100 steps.

#### 4.1.3 ENTROPY FOR ANALYZING ATTENTION CONCENTRATION EFFECTS

Given the attention matrix $\mathbf{A} \in \mathbb{R}^{l \times l}$ at one layer, we use entropy $e$ to represent its concentration effect. To obtain entropy $\boldsymbol{E}$, calculate the importance vector $\mathbf{a} = \frac{1}{l} \sum_{j=1}^{l} A_{ij}$ firstly where $\mathbf{A}$ is a lower triangular matrix. The entropy can be formulated as follows:

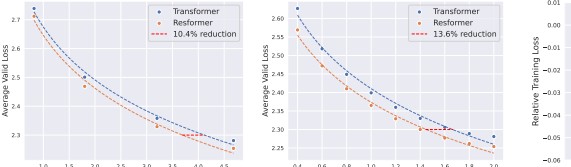 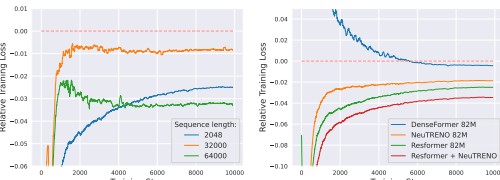

Figure 6: (Left) Validation loss as model size scales from 82M to 468M parameters. (Right) Validation loss for the 468M parameter model evaluated every 2B tokens. ResFormer achieves approximately 10.4%-13.6% reduction in both model parameters and training data.

Figure 7: (Left) The relative training curve between a 82M ResFormer and Transformer across different training sequence lengths. (Right) Relative training loss of various Transformer variants compared to the vanilla Transformer model, with model size fixed at 82M parameters.

$$\boldsymbol{e} = -\sum_{i=1}^{l} \boldsymbol{a}_i' \log \boldsymbol{a}_i'. \tag{11}$$

where $a_i'^i = a_i / \left( \sum_{i=1}^{l} a_i \right)$ for $i = 1, 2, \ldots, l$ and the higher the entropy $\boldsymbol{e}$, the greater the degree of clustering in $\boldsymbol{a}$, i.e., attention matrix $\mathbf{A}$ is more likely to focus on several specific tokens.

### 4.1.4 SPECTRAL DECOMPOSITION FOR ANALYZING REPRESENTATIONS

Spectral Decomposition is a classical method to analyze the representations of models. Zhu et al. (2021) suggests that the eigenvectors with larger eigenvalues are more transferable. Here we use spectral decomposition to analyze the feature space of value $\boldsymbol{v}$ of one layer as following:

$$\frac{1}{l} \sum_{i=1}^{l} \boldsymbol{v}_i \boldsymbol{v}_i^T = \sum_{j=1}^{d} \boldsymbol{w}_j \boldsymbol{\lambda}_j \boldsymbol{w}_j^T. \tag{12}$$

where $\boldsymbol{w}_j$ is the $j$-th eigenvector with eigenvalue $\boldsymbol{\lambda}_j$ for $j = 1, 2, \ldots, d$ and $d$ is the dimensionality of the value's feature space.

### 4.2 RESFORMER vs. VANILLA TRANSFORMER

We trained ResFormer and vanilla Transformer with different model size on data with different sequence lengths. In Fig. 7 (Left), ResFormer consistently outperforms vanilla Transformer throughout training across different training sequence lengths. Additionally, the results in Fig. 7 (Right) illustrate that ResFormer outperforms DenseFormer and NeuTRENO. Furthermore, integrating ResFormer with NeuTRENO leads to additional performance improvements.

We also analyzed how ResFormer and Transformer scale at model size and data size. ResFormer and Transformer are trained on similar experiment setting. On the one hand, we trained model with 82M, 180M, 320M and 468M parameters on 20B training tokens and evaluated them on a separate validation set. As shown in Fig.6 (Left), ResFormer achieves equivalent validation loss to the Transformer while utilizing 10.4% fewer model parameters. On the other hand, we evaluated the 468M models every 2B tokens and ResFormer needs 13.6% fewer training tokens to achieve the same validation loss as Transformer. The validation loss for these models is available in Appendix A.6.

We further test the variant of ResFormer defined as $\mathbf{U}_n = \mathbf{A}_n(\mathbf{V}_n + \lambda \mathbf{V}_1)$. As shown in Fig.8, ResFormer can accommodate a wide range of $\lambda$ values and the performance improves as $\lambda$ increases, achieving the best results at $\lambda = 2$. Regardless of the value of $\lambda$, ResFormer consistently outperforms Transformers. It suggests that the success of ResFormer lies in the use of $\mathbf{V}_1$ and the mapping by $\mathbf{A}_n$. The ablation study of different hyperparameters $\lambda$ for NeuTRENO, as defined in Equation 8, can be found in the Appendix A.3.

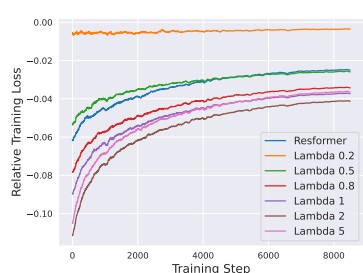

Figure 8: Ablation study of different $\lambda$ for ResFormer.

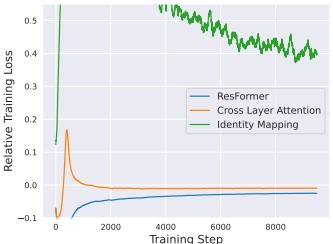 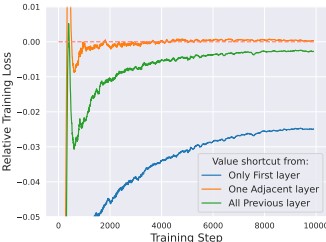

Figure 9: Ablation study of adding residual connection to queries or keys.

Figure 10: Ablation study of adding residual connection using different mapping matrix.

Figure 11: Ablation studies on which historical layer's value to include in residual connections.

### 4.3 ABLATION STUDY OF RESIDUAL CONNECTION

In Eqn. 4, we employ residual connections for the values. We compare this approach with models that add residual connections to queries or keys. The results, shown in Fig. 9, indicate that only residual connections for values yield positive effects. One possible explanation is that attention mechanisms are sensitive to perturbations, and modifying queries or keys significantly impacts it.

Moreover, we compare with the models based on Eqn. 2 and Eqn. 6. The results in Fig. 10 align with Fig. 4, showing that identity mapping causes significant perturbations, leading to poor performance. Interestingly, ResFormer achieves an even lower final loss than ResFormer. It suggests that ResFormer's impact on the attention optimization is better by mitigating value-state drains.

When determining the mapping method and target value, it is crucial to consider which historical layers' values should be included in the residual connection. Fig. 11 shows that each Transformer layer should add a shortcut to the first layer's value rather than to the nearest preceding layer or all previous layers, highlighting the first-layer value's critical importance. A potential explanation is that incorporating values from other layers may dilute the impact of the first-layer value.

### 4.4 DOWNSTREAM EVALUATIONS

We compare the different models on several classical reasoning tasks following (Zhang et al., 2024) in a zero-shot way. The tasks include Hellaswag (Zellers et al., 2019), OpenBookQA (Mihaylov et al., 2018), WinoGrande (Sakaguchi et al., 2019), ARC-Easy and ARC-Challenge (Clark et al., 2018) and PIQA (Bisk et al., 2020). The results in Table 1 show that ResFormer achieved an average accuracy improvement of nearly 3% compared to the vanilla Transformer.

| Model | Max Length | HellaSwag | Obqa | WinoGrande | ARC-c | ARC-e | PIQA | Avg |
|---|---|---|---|---|---|---|---|---|
| Transformer | 2,048 | 0.263 | 0.142 | 0.492 | 0.199 | 0.331 | 0.572 | 0.333 |
| ResFormer | 2,048 | 0.273 | 0.148 | 0.512 | 0.182 | 0.414 | 0.604 | 0.355 |
| Transformer | 64,000 | 0.267 | 0.142 | 0.485 | 0.179 | 0.322 | 0.570 | 0.328 |
| ResFormer | 64,000 | 0.274 | 0.136 | 0.513 | 0.184 | 0.407 | 0.588 | 0.350 |

Table 1: Zero-shot accuracy on commonsense reasoning tasks.

### 4.5 VISUALIZATION OF RESFORMER

To figure out why ResFormer can achieve better performance on language modeling tasks than vanilla Transformer, we conduct visualization based on the eigenvalue decomposition discussed in Section 4.1.4. After sorting the eigenvalues in descending order, we compute the average eigenvalue for each layer across 1,000 randomly sampled pre-train data examples. The results in Fig. 12 indicate that the value states generated by most layers of the ResFormer exhibit stronger representational capacity compared to those of the vanilla Transformer.

We also analyze the attention concentration effects mentioned in Section 4.1.3 using the same batch of test data. Fig. 1 (Right) illustrates that the clustering effect of attention increases significantly

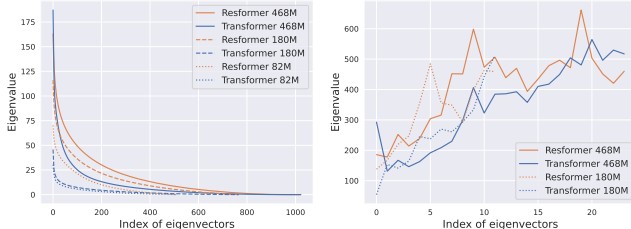

Figure 12: Left: Distribution of eigenvalues for the value vectors in the first layer of ResFormer and Transformer. Right: Maximum eigenvalue for each layer of ResFormer and Transformer.

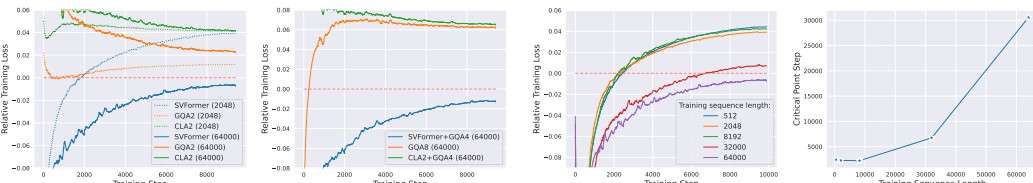

Figure 13: The relative training loss for SV-Former and other $KV$ efficient model compared with vanilla attention. The numbers in parentheses represent the training sequence length. Left: Model with nearly $1/2$ $KV$ cache. Right: Model with nearly $1/8$ $KV$ cache.

Figure 14: Left: The relative training loss for SV-Former under different sequence lengths with a fixed batch size of 2M tokens. Right: Analysis of critical point, and we predict it for length 64,000 using linear regression with the last 1,000 data points.

with the number of layers for the vanilla Transformer, whereas the clustering effect is relatively less pronounced for the ResFormer. We further visualize the attention weights, value-state norms $\|\boldsymbol{v}\|_2$, and hidden-state norms $\|\boldsymbol{h}\|_2$ of tokens at different layers and positions, with detailed results in Appendix A.2. Given that attention clustering often occurs on the first token, we primarily show its results in Fig. 2. The results indicate that using ResFormer significantly mitigates attention sinks (Xiao et al., 2024), value-state drains (Guo et al., 2024b) and residual-state peaks (Sun et al., 2024). Guo et al. (2024a) attributes these phenomena to the mutual reinforcement mechanism of model and we suggest that the value shortcut disrupts this mechanism by alleviating value-state drains. Specifically, for tokens lacking semantic information like start tokens, a large value state magnitude can adversely affect the prediction of subsequent tokens if they are overly attended to. When there is no value-state drains, models will reduce attention clustering to these tokens to minimize loss.

### 4.6 SVFORMER *vs.* GQA

In the Fig. 13, at a training sequence length of 64,000, SVFormer demonstrates lower final loss compared to existing $KV$-efficient methods such as CLA and GQA. Moreover, it can be used concurrently with GQA to enhance $KV$ efficiency further. However, we observed that with a training sequence length of 2,048, SVFormer underperforms compared to GQA. The results indicate that sequence length significantly affects SVFormer's performance. Thus, we conducted more comprehensive experiments on sequence length.

Results in Fig. 14 (Left) demonstrate that SVFormer will always be gradually surpassed by vanilla attention during training while its training speed is faster than vanilla Transformer at the early stage. However, as the training sequence length increases, the SVFormer model performs better. In this way, we focus on the critical point, defined as the number of training steps exceeded. Fig. 14 (Right) illustrates that the relationship between the critical point and sequence length exhibits an exponential trend. We argue that it's due to the challenge deep models face in fully optimizing the increasingly larger first-layer value matrix as the training sequence length grows.

### 4.7 OTHER FACTORS INFLUENCING SVFORMER

Intuitively, the training effectiveness of SVFormer is influenced by factors such as the maximum learning rate, warmup steps, model size, and other factors beyond just the training sequence length. We conducted experiments to explore these relationships.

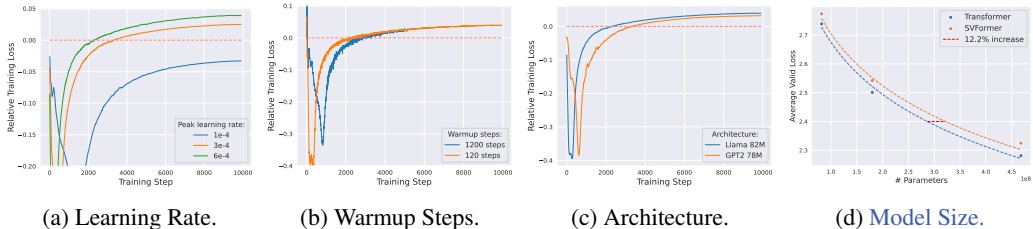

| (a) Learning Rate. | (b) Warmup Steps. | (c) Architecture. | (d) Model Size. |

Figure 15: The relative training loss for SVFormer under different hyper-parameter setting and the validation loss as model size scales from 82M to 468M parameters.

Based on the results shown in Fig. 15a and Fig. 15b, a smaller learning rate benefits SVFormer more, with warmup's impact being comparatively small. This could be attributed to the model's outcomes being closely tied to the total summed learning rate, which has weak connection with warmup steps (Kaplan et al., 2020). Moreover, larger models often require smaller learning rates to ensure training stability, making them more suitable for using SVFormer.

Llama-like models and GPT2-like models exhibit similar critical points and final losses (see Fig. 15c). This suggests that the difference between SVFormer and the vanilla Transformer is not sensitive to architecture. Compared with Transformer, SVFormer requires a 12.2% increase in parameters to achieve the same validation loss while reducing the $KV$-cache by nearly half.

## 4.8 ABLATION STUDY OF SVFORMER

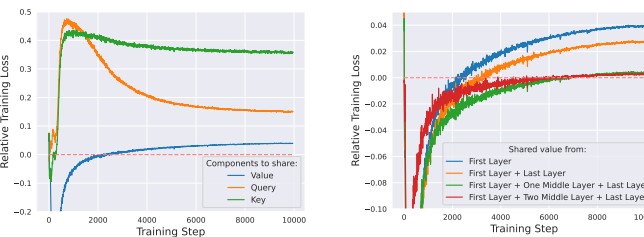

Figure 16: Ablation study of sharing first layer's query(key) across all layers.

Figure 17: Ablation study on sharing values from different numbers of layers.

To better understand SVFormer, we conduct several ablation experiments. We first observe the effects of sharing the first layer's queries or keys across all layers in Fig. 16, finding that this significantly impacts model performance, similar to the results in Fig. 5. Additionally, sharing the first layer's values in a multi-layer network may reduce the network's "effective depth." By updating the shared values using intermediate layers as "anchors," we find that increasing the number of "anchors" improves performance, as shown in Fig. 17.

## 5 CONCLUSION

In this paper, we propose the concept of attention concentration, a problem that arises from stacking multiple attention layers. From the perspective of cross-layer attention, we derive ResFormer, which adds a residual connection between the value vectors of the current layer and those of the first layer before the attention operation to alleviate attention concentration. Additionally, we introduce SVFormer, based on ResFormer, which reduces the $KV$ cache by nearly half. We conducted comprehensive experiments on the language modeling task to validate the advantages of these two Transformer variants in different scenarios.

## ETHICS STATEMENT

On the one hand, the data employed in this paper is sourced from publicly available datasets provided by the company, which have undergone a certain level of filtering. On the other hand, the models trained in our study are solely utilized for experimental analysis and will not be publicly deployed.

## REPRODUCIBILITY STATEMENT

We have detailed the complete experiments setup such as batch size, optimizer, learning rates in Section 4.1.1. Besides, we will release source codes once our paper is made public. These resources should be sufficient to reproduce results of the paper.

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

## A   APPENDIX

### A.1   TOKEN SIMILARITY ANALYSIS

Attention concentration tends to make embeddings of different tokens more similar, resulting in over-smoothing. The extent of over-smoothing can be assessed by calculating the average token similarity $s$ of the hidden states using the following formula:

$$s = \frac{1}{l(l-1)} \sum_{i=1}^{l} \sum_{j=1,j\neq i}^{l} \mathrm{Sim}\left(\boldsymbol{h}_i, \boldsymbol{h}_j\right). \tag{13}$$

where $\{\boldsymbol{h}_i\}_{i=1}^{l}$ is the hidden state of the $i$-th token and $\mathrm{Sim}(\cdot)$ denotes the operation of cosine similarity. The results in Fig. 18 are align with the results in Fig. 1. In the case of Llama and Mistral, the average token similarity demonstrates an "M"-shaped pattern with increasing network depth, while entropy follows a "W"-shaped pattern at corresponding positions. These trends indicate a strong correlation between attention concentration and over-smoothing.

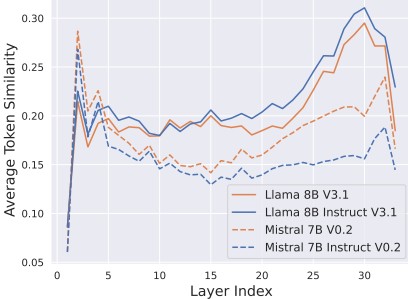

Figure 18: The average token similarity of hidden states across layers in Llama and Mistral.

### A.2   ATTENTION CONCENTRATION VISUALIZATION

We visualize the token importance, norms of value states and norms of hidden states for tokens at different position across layers. The results are averaged from 1,000 different sequences so that only

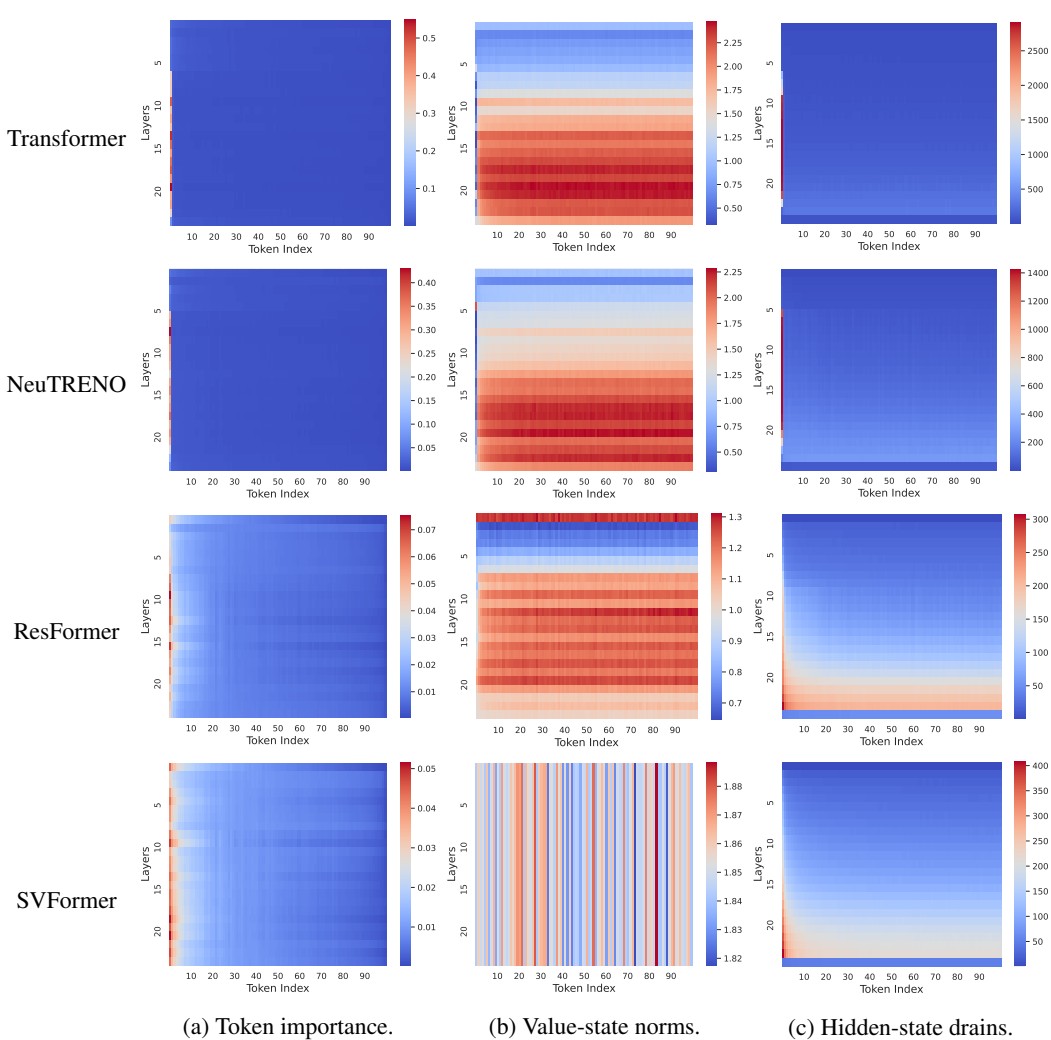

(a) Token importance.  (b) Value-state norms.  (c) Hidden-state drains.

Figure 19: Visualization of token importance, value state norms, and hidden state norms across different token positions and layers in 468M models.

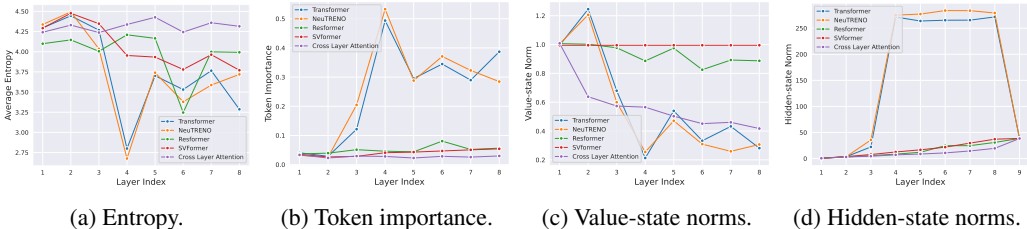

(a) Entropy.      (b) Token importance.      (c) Value-state norms.      (d) Hidden-state norms.

Figure 20: Token analysis in 82M models. (a) Importance entropy of sequences; (b) Token importance, (c) value-state norms, and (d) hidden-state norms of the first token across layers.

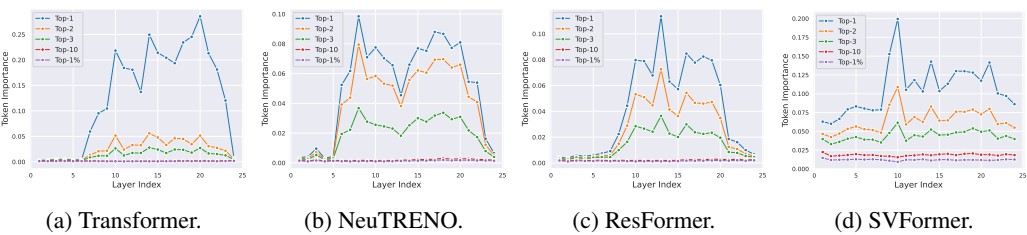

(a) Transformer.      (b) NeuTRENO.      (c) ResFormer.      (d) SVFormer.

Figure 21: The distribution of token importance for different models at different layers.

the start token is the same and special across all sequences. Fig. 19 (First column) demonstrates that the start token easily attracts massive attention despite lacking semantic information for Transformer and NeuTRENO. For ResFormer, the importance of the start token is less than 10 times that of tokens at other positions, indicating that tokens carrying semantic information receive more attention. Moreover, both Transformer and NeuTRENO exhibit significant value-state drains (Guo et al., 2024b) and residual-state peaks (Guo et al., 2024a; Sun et al., 2024) on the start token at certain layers. In contrast, for ResFormer, the value state norm of the start token exceeds half the magnitude of other tokens, while the peak hidden state norm is less than triple the average. Fig.21 further illustrates the distribution of token importance, where TOP-$i$ represents the $i$-th largest token importance within a sequence. Compared to Transformer and NeuTRENO, ResFormer and SVFormer exhibit a more uniform distribution of token importance.

Similar to Fig.2, we conducted experiments on 82M models, with results shown in Fig.20. We also illustrate the attention pattern of cross-layer attention introduced in Eqn. 2. The results demonstrate that while cross-layer attention successfully mitigates the problem of attention concentration, it still exhibits value-state drains.

## A.3 ABLATION STUDY OF NEUTRENO

NeuTRENO is sensitive to the choice of hyperparameter $\lambda$ which is task-dependent. In the appendix of Nguyen et al. (2023), it is reported that $\lambda$ is set to 0.6 for image classification and segmentation tasks, and 0.4 for language modeling tasks. Fig. 22 indicates that $\lambda = 0.4$ achieves the best results in our training dataset so that we choose $\lambda = 0.4$ for comparison. Besides, we empirically choose $\lambda = 0.2$ for NeuTRENO when combined with ResFormer.

## A.4 PRE-TRAIN DATASET

Based on the equation $D \geq 5000 \cdot N^{0.74}$ (Kaplan et al., 2020) where $D$ is data size and $N$ is the number of non-embedding parameters, we need to collect at least 17.5B for model has N = 700M non-embedding parameters (corresponding to complete 1B model with 2,048 hidden size, 50,277 vocab size and 2,048 sequence length) to avoid over-fitting. Besides, Xie et al. (2024) indicates that the mixture proportions of pre-training data domains significantly affects the training results. In this way, we sampled 20B tokens data from original 627B data based on the original data proportions shown in the Table 2.

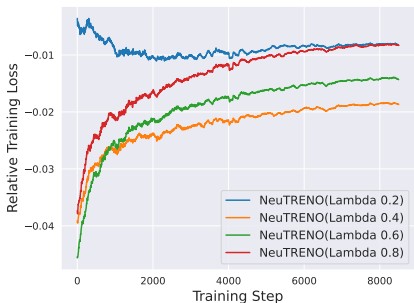

Figure 22: Ablation study of different $\lambda$ for NeuTRENO.

| Data source | proportions | Tokens |
|---|---|---|
| Commoncrawl | 50% | 10 B |
| C4 | 20% | 4 B |
| GitHub | 10% | 2 B |
| Books | 5% | 1 B |
| ArXiv | 5% | 1 B |
| Wikpedia | 5% | 1 B |
| StackExchange | 5% | 1 B |

Table 2: The details of pre-train dataset.

## A.5 TRAINING DETAILS

| Max Sequence Length | 512 | 2,048 | 8,192 | 32,000 | 64,000 |
|---|---|---|---|---|---|
| Total Batch Size | 4,096 | 1,024 | 256 | 64 | 32 |
| Per-GPU Batch Size | 128 | 32 | 8 | 2 | 1 |
| Gradient Accumulation Step | | | 32 | | |
| GPUs | | | 8 | | |

Table 3: Training details for training dataset with different sequence length.

Section 4.1.1 introduces the main experimental hyperparameters used in the paper. This section further details the training parameters for various model sizes and training sequence lengths. Table 4 demonstrates the differences among models of various sizes. The configurations for the number of layers, attention heads, hidden dimensions, and FFN dimensions are based on Biderman et al. (2023). Additionally, the $\lambda$ in Eqn. 8 is set to be 0.4 for NeuTRENO. Moreover, as reported in Table 3, the batch size that a single GPU can accommodate varies depending on the length of the training sequences. Note that the total number of tokens in each batch is consistently 2 million.

## A.6 VALIDATION LOSS ON SLIMPAJAMA

Section 4.1.2 introduces to use relative training loss as a main evaluation matrix. Table 5 reports the validation loss for differnt model on the whole validation split of slimpajama.

| Model Size | 2M | 82M | 180M | 468M |
|---|---|---|---|---|
| Layers | 4 | 8 | 12 | 24 |
| Attention Heads | 2 | 8 | 12 | 16 |
| Hidden Dimension | 16 | 512 | 768 | 1,024 |
| FFN Dimension | 56 | 1,792 | 2,688 | 3,584 |
| Tie Word Embedding | False | | | |
| (Peak Learning Rate, Final Learning Rate) | $(6e-4, 6e-5)$ | | | |
| Learning Rate Schedule | Cosine Decay | | | |
| Vocabulary Size | 50,277 | | | |
| Activation Function | SwiGLU | | | |
| Position Embedding | RoPE ($\theta = 10,000$) | | | |
| Batch Size | 2M tokens | | | |
| Data Size | 20B tokens | | | |
| (Warmup Steps, Training Steps) | (120, 10,000) | | | |
| Adam $\beta$ | (0.9, 0.95) | | | |
| Dropout | 0.0 | | | |
| Weight Decay | 0.1 | | | |

Table 4: Training details for models with different size.

| Model | Common Crawl | C4 | Github | Stack Exchange | Wikipedia | Book | Arxiv | Avg. |
|---|---|---|---|---|---|---|---|---|
| Transformer (82M) | 3.3595 | 3.5388 | 1.4247 | 2.3872 | 2.9047 | 3.3797 | 2.1779 | 2.7389 |
| Transformer (180M) | 3.0961 | 3.2834 | 1.2451 | 2.1651 | 2.5897 | 3.1309 | 2.0001 | 2.5015 |
| Transformer (468M) | 2.8514 | 3.0430 | 1.0908 | 1.9628 | 2.2821 | 2.8979 | 1.8362 | 2.2806 |
| ResFormer (82M) | 3.3362 | 3.5191 | 1.3941 | 2.3592 | 2.8646 | 3.3572 | 2.1518 | 2.7117 |
| ResFormer (180M) | 3.0631 | 3.2504 | 1.2200 | 2.1350 | 2.5435 | 3.0994 | 1.9732 | 2.4692 |
| ResFormer (468M) | 2.8214 | 3.0115 | 1.0730 | 1.9388 | 2.2477 | 2.8696 | 1.8142 | 2.2537 |

Table 5: Validation loss on slimpajama.

