# OpenReview forum: "Value Residual Learning For Alleviating  Attention Concentration In Transformers"
_ICLR.cc/2025/Conference — Submitted to ICLR 2025_

### Official Review · Reviewer_dQDC · 2024-11-01

**Soundness:** 2
**Presentation:** 1
**Contribution:** 2
**Rating:** 3
**Confidence:** 4

**Summary:**

This paper presents ResFormer and SVFormer, two Transformer model variants that address challenges with deep Transformers, particularly attention concentration where deeper layers focus too much on fewer tokens. ResFormer incorporates a residual connection from the initial layer's values to subsequent layers, thus approximating cross-layer attention without heavy computational costs. SVFormer simplifies further by sharing the value embedding from the first layer across all layers, reducing memory usage by nearly half and accelerating training.

**Strengths:**

1. The paper offers an interesting twist on standard residual connections by applying them specifically to V instead of the usual hidden state H. This approach targets the common issues of over-smoothing and information loss in deep Transformers.

2. SVFormer aims to make Transformer models more efficient by sharing the same value embeddings across layers, reducing memory and computation needs. This design could help make large models faster and more practical for applications with long sequences.

**Weaknesses:**

1. **Problem Definition and Motivation**: The problem of "attention concentration" is not clearly defined or sufficiently justified. It is essential for the authors to establish a precise understanding of this issue and clarify why it is a significant challenge within Transformer architectures. Without a thorough introduction and motivation for addressing "attention concentration," it remains unclear what gap this work aims to fill, and the importance of resolving it is left ambiguous.

2. **Novelty and Theoretical Basis**: The proposed approach largely resembles existing residual connections in Transformers, as seen in architectures like ViT and LLaMA. The primary difference with ResFormer appears to be the application of residuals to the value V alone, rather than to the hidden state H as in traditional models. However, this adjustment lacks theoretical grounding and rigorous analysis, especially with regard to the SVFormer, which further simplifies by removing layer-specific values. This simplification seems ad-hoc and trivial, as no theoretical guarantees or insights are offered to support the effectiveness or necessity of such changes.

3. **Experimental Setup and Comparisons**: The experiments are limited and do not provide a thorough benchmark. Although the models are trained on a LLaMA-like architecture, there is no comparative performance evaluation against other prominent Transformer-based or SSM-based models. Furthermore, there are no tests involving visual downstream tasks, which would have strengthened the claims of improvement in Transformers and provided a more comprehensive evaluation across different modalities, especially for encoder-only tasks.

4. **Evaluation of Attention Mechanisms**: An essential part of evaluating any modification to Transformer architectures is understanding how the attention patterns differ from those in the vanilla Transformer. Although the paper discusses attention concentration, it does not provide visualizations or statistical analysis of the multi-head attention weights to demonstrate the proposed method's effect on attention distribution. Such an investigation is critical for validating the claims and understanding how the modifications impact attention dynamics.

**Questions:**

same as weakness

---

> ### Author Response · Authors · 2024-11-24
> **Response to Reviewer dQDC (1/4)**
>
> Thank you for your thoughtful review and suggestions. We have revised the manuscript's presentation based on your comprehensive feedback. We address the concerns below:
>
> ### **Q1: Problem Definition and Motivation**
> We clarify three important concepts in language models and their relationships:
>
> #### **Key concepts:**
> 1. Over-smoothing: As the model's depth increases, token representations become increasingly similar.
> 2. Attention sink [1]: Language models allocate substantial attention to low-semantic tokens, particularly the start token.
> 3. Attention concentration: The model's attention focuses on fewer tokens as network depth increases.
>
> #### **Relationships:**
>
> 1. Over-smoothing vs. Attention concentration: Over-smoothing is inherent to model architecture, while attention concentration emerges during training. Figure 1 (Left) demonstrates that randomly initialized models exhibit over-smoothing but not attention concentration. These two phenomena interact, as shown in Figure 1 (Middle) and Appendix A.1 Figure 18.
> 2. Attention sink vs. Attention concentration: Attention sink is a specific manifestation of attention concentration in certain layers, while attention concentration describes the changing attention patterns as information propagates through layers. Deep models exhibit a "concentration - dispersion - concentration" attention pattern, as illustrated in Figure 1 (Middle).
>
> #### **Reasons for alleviating attention concentration:**
>
> On the one hand, the concentrated phases may lead to potential information loss. On the other hand, mitigating attention concentration could result in more interpretable attention maps and potentially improve downstream task performance [2][3].
>
> [1] Guangxuan Xiao, et al. " Efficient streaming language models with attention sinks." ICLR (2024).
>
> [2] Timothee Darcet, et al. "Vision transformers need registers." ICLR (2024).
>
> [3] Mingjie Sun, et al. "Massive activations in large language models." arXiv (2024).

---

> ### Author Response · Authors · 2024-11-24
> **Response to Reviewer dQDC (2/4)**
>
> ### **Q2: Novelty and Theoretical Basis**
> The effectiveness of Resformer and SVformer can be attributed to three key factors:
>
> 1. The application of shortcuts to the value component
> 2. The utilization of information from the first layer
> 3. The implementation of shortcuts before each layer's attention operation
>
> Our findings support these claims:
>
> - Figures 9 and 16 demonstrate the necessity of applying shortcuts to the value-states.
> - Figure 11 illustrates the importance of using the value from the first layer.
> - Figure 10 shows that when the identity map (equivalent to applying the shortcut after the attention operation) is used for residual value, the performance decreases significantly.
>
> #### **Theoretical grounding and analysis:**
>
> Attention concentration typically emerges after the second or third network layer [1] [2] [3] and is associated with value-state drains (decreased magnitude of value states) [1], and hidden-state peaks (increased magnitude of hidden states) [2]. [3] shows a mutual reinforcement mechanism exists between value-state drains and attention concentration.
>
> The first layer always shows no attention concentration, and our methods leverage this characteristic. ResFormer indirectly mitigates attention concentration. It leverages the absence of value-state drains in the first layer by introducing a value residual connection. This alleviates value-state drains in deeper layers, thereby disrupting the mutual reinforcement between attention concentration and value-state drains, as shown in Figure 1 (Right) and Figure 2. Specifically, for tokens lacking semantic information, such as the first token, a large value state magnitude can adversely affect the prediction of future tokens if it is overly attended to, resulting in higher loss. Besides, SVformer can also alleviate attention concentration for the same reasons.
>
> #### **Novelty:**
>
> Recent studies have linked attention sink to implicit biases during pretraining, with most existing solutions focusing on additional tokens (registers) [4] or additional keys and values (explicit attention bias) [2]. While these methods redirect attention concentration from tokens within the sequence to additionally introduced bias terms, Resformer and SVformer effectively utilize the inherent absence of attention concentration in the network's first layer. This strategy provides a simple yet efficient means of mitigating attention concentration.
>
> In language modeling tasks, Resformer leverages the first-layer values already stored in the KV cache or intermediate activations to achieve significant performance improvements across multiple settings. This enhancement is accomplished almost without additional memory overhead and computational cost (involving only several matrix addition operations).
>
> Building on CLA [5], we found that sharing values across layers is more useful than sharing both keys and values. SVFormer reduces KV cache size by nearly half through value sharing across all layers, with only a small performance penalty. SVFormer's effectiveness improves with increased learning rates and training sequence lengths, outperforming other KV-efficient methods like GQA and CLA in these scenarios.
>
>
> [1] Zhiyu Guo, et al. "Attention score is not all you need for token importance indicator in kv cache reduction: Value also matters." Arxiv(2024).
>
> [2] Mingjie Sun, et al. "Massive activations in large language models." ArXiv (2024).
>
> [3] Tianyu Guo, et al. "Active-dormant attention heads: Mechanistically demystifying extreme-token phenomena in llms." ArXiv (2024).
>
> [4] Timothee Darcet, et al. "Vision transformers need registers." ICLR (2024).
>
> [5] William Brandon, et al. "Reducing transformer key-value cache size with cross-layer attention." ArXiv (2024).

---

> ### Author Response · Authors · 2024-11-24
> **Response to Reviewer dQDC (3/4)**
>
> ### **Q3: Experimental Setup and Comparisons**
> #### **Loss-based experiments:**
> Perplexity (Loss) is a crucial metric for evaluating language models during pretraining [1]. We conducted extensive experiments using Perplexity as the primary indicator to demonstrate ResFormer's superior performance compared to Transformer and its variants across various model sizes, training token counts, and sequence lengths. Through ablation studies, we explored ResFormer's ability to achieve lower Perplexity.
>
> For comparing KV-efficient methods, we adopted the approach from [2], combining different techniques and evaluating Perplexity under various KV-cache reduction levels. Our experiments consistently show that SVFormer achieves 50% KV cache reduction with minimal loss penalty across different learning rate schedules, model sizes, data sizes, training sequence lengths, and architectures.
>
> **Added content:**
>
> 1. Our experimental setup was informed by several previous studies [3] [4]. We have provided detailed hyperparameters for models of various sizes and different training sequence lengths in Appendix A.5, Tables 3 and 4.
> 2. To further assess the model's robustness, we tested variants of ResFormer defined as $U_n = A_n(V_n + λV_1)$ and NeuTRENO defined as $U_n = A_nV_n + λ(V_1 - V_n)$ with different $λ$ values. The results are shown in Figure 8 and Appendix A.3 Table 22 respectively. Our results indicate that ResFormer can accommodate a wider range of $λ$ values compared to NeuTRENO. Specifically, ResFormer achieves optimal performance at $λ=2$, whereas NeuTRENO's performance peaks at $λ=0.4$.
> 3. As mentioned in Section 4.1.2, we trained our models for only one epoch on the training dataset. Consequently, the training loss on each new batch can be considered equivalent to validation loss. In Appendix A.6 Table 5, we provide a detailed report of the loss for models of different sizes on various domain-specific validation sets. These validation sets, provided officially, do not overlap with the training data. Additionally, the scaling curves in Figure 6 are also based on validation losses.
>
> #### **Downstream tasks evaluation:**
> We evaluated the trained ResFormer and Transformer models on several classical downstream tasks following [5] (see Table 1). Our best-performing models required 2-4 days of training time on 8 A100 GPUs. Due to the scale limitations of our models, we did not evaluate them on more challenging benchmarks such as MMLU [6].
>
> **Added content:**
> Following your suggestion, we conducted object classification experiments on ImageNet-1K using the DeiT-base model [7]. We adhered to the hyperparameters specified in the original paper and trained for 300 epochs. Results indicate that ResFormer exhibits faster initial training but encounters overfitting in later stages. This overfitting was not observed in language modeling tasks, where models do not overfit most of the time. These findings suggest that for vision recognition tasks, ResFormer may require additional regularization techniques to optimize performance.
>
> | Metric | Model | Epoch 50 | Epoch 100 | Epoch 150 | Epoch 200 | Epoch 250 | Epoch 300 |
> |--------|-------|---------|---------|---------|---------|---------|---------|
> | Test Acc@1 | DeiT | 62.53 | 71.84 | 75.77 | 78.89 | 81.03 | 81.99 |
> |  | DeiT-NeuTRENO | 61.83 | 72.09 | 75.89 | 78.85 | 80.88 | 81.64 |
> |  | DeiT-ResFormer | 66.23 | 73.03 | 76.44 | 79.25 | 80.83 | 81.59 |
> | Train Acc@5 | DeiT-Transformer | 85.11 | 91.09 | 93.11 | 94.57 | 95.42 | 95.68 |
> |  | DeiT-NeuTRENO | 84.65 | 91.03 | 93.13 | 94.56 | 95.4 | 95.59 |
> |  | DeiT-ResFormer |87.63 | 91.67 | 93.41 | 94.54 | 95.12 | 95.58 |
> | Test Loss | DeiT-Transformer | 1.74 | 1.26 | 1.07 | 0.94 | 0.85 | 0.82 |
> |  | DeiT-NeuTRENO | 1.74 | 1.23 | 1.05 | 0.93 | 0.85 | 0.83 |
> |  | DeiT-ResFormer | 1.54 | 1.21 | 1.04 | 0.93 | 0.88 | 0.89 |
> | Train Loss | DeiT-Transformer | 4.32 | 3.71 | 3.4 | 3.1 | 2.78 | 2.6 |
> |  | DeiT-NeuTRENO | 4.32 | 3.7 | 3.41 | 3.13 | 2.82 | 2.65 |
> |  | DeiT-ResFormer | 4.05 | 3.58 | 3.3 | 3.0 | 2.67 | 2.48 |
>
> [1] Jared Kaplan, et al. "Scaling Laws for Neural Language Models." Arxiv(2020).
>
> [2] William Brandon, et al. "Reducing transformer key-value cache size with cross-layer attention." ArXiv (2024).
>
> [3] Albert Gu, et al. "Mamba: Linear-time sequence modeling with selective state spaces." COLM(2024).
>
> [4] Stella Biderman, et al. "Pythia: A suite for analyzing large language models across training and scaling." ICML(2023).
>
> [5] Haoyi Wu, et al. "Layer-Condensed KV Cache for Efficient Inference of Large Language Models." ACL(2024).
>
> [6] Dan Hendrycks, et al. "Measuring Massive Multitask Language Understanding." ICLR(2021).
>
> [7] Hugo Touvron, et al. "Training data-efficient image transformers & distillation through attention." ICML(2021).

---

> ### Author Response · Authors · 2024-11-24
> **Response to Reviewer dQDC (4/4)**
>
> ### **Q4: Evaluation of Attention Mechanisms**
>
> We add several experiments to visualize the attention patterns in different models. We visualize attention weights, value-state norms, and hidden-state norms of tokens at different layers and positions, with detailed results in Appendix A.2 Figure 19. Given that attention clustering often occurs on the first token, we separately show its results in Figure 2 and Figure 20. Besides, we illustrate the distribution of token importance for different models at different layers in Appendix A.2 Figure 21. The results indicate that Resformer and SVformer largely alleviates the problem of attention sinks, value-state drains and hidden-state peaks. And they also exhibit a more uniform distribution of token importance wchich represents the distribution of attention weights.

---

### Official Review · Reviewer_nCnK · 2024-11-01

**Soundness:** 2
**Presentation:** 2
**Contribution:** 3
**Rating:** 3
**Confidence:** 3

**Summary:**

This paper studies the problem of attention concentration in Transformers and proposes solutions that try to approximate cross-layer attention by incorporating the "value" from first layer into subsequent layers. There are two solutions: ResFormer that uses residual mapping and SVFormer that uses the same V across all layers. Experiments show that the proposed solutions perform better than baselines on language modeling tasks.

**Strengths:**

- The paper introduces a relatively new and important problem that affects existing Transformer architecture. This is useful towards understanding the dynamics and behavior of Transformers.
- The proposed solutions only require small changes to existing Transformer architecture. They can be immediately useful for many existing Transformer-based models.
- The paper provides a good analysis and ablation study on ResFormer and SVFormer that demonstrate their benefits over existing Transformer. Particularly, ResFormer is shown to be achieving higher token importance entropy (i.e., less attention concentration) than traditional Transformer.

**Weaknesses:**

- The authors claim that cross-layer attention is useful at reducing the effect of attention concentration but it is unclear why this would be the case. This work is built on the premise that ResFormer approximates cross-layer attention and thus it is effective against attention concentration. But we do not really know that cross-layer attention provides such a benefit. The author should perform some analysis and/or small-scale experiment on a baseline that actually uses cross-layer attention to check its behavior against that of ResFormer.

- It is hard to disentangle the effects from: (1) reducing attention concentration; (2) ease of optimization in the proposed solutions. Using V in the form of residual mapping (ResFormer) or layer sharing (SVFormer) should make it easier to optimize network parameters during training. It may be possible that the accuracy improvements are largely attributed to the ease of optimization rather than attention concentration reduction. The authors should explain this.

- It would also be interesting to see how well the proposed methods work for non-language tasks and architectures like ViT (image recognition).

**Questions:**

- What are the reasons of using LLama-like architecture and SlimPajama dataset?

---

> ### Author Response · Authors · 2024-11-24
> **Response to Reviewer nCnK**
>
> Thank you for your constructive review and appreciation of our work!
>
> ### **Q1: The effect of cross-layer attention on attention concentration.**
> The visualization results in Appendix Figure 20 can support that cross-layer attention alleviates attention concentration. Attention concentration occurs when the angles between certain tokens' keys and other tokens' queries are small [1], often accompanied by low value-state norms for these tokens [2]. As shown in Figure 1 and Figure 2, the first layer typically does not exhibit attention concentration, and its key-value pairs do not display the aforementioned characteristics. This suggests that applying cross-layer attention using the first layer's key-value pairs could enable the model to maintain appropriate attention distribution across important tokens throughout the entire sequence, even in deeper layers.
>
> In Figure 10, the ``Gold Attention" actually corresponds to the model using cross-layer attention on the first layer. We have revised this potentially misleading representation in the manuscript. Results indicate that cross-layer attention underperforms compared to ResFormer. This is because cross-layer attention does not directly affect the current layer's tokens, which still experience value-state drains, as shown in Figure 20. In contrast, ResFormer's impact on the attention optimization process is more fundamental through mitigating value-state drains.
>
> ### **Q2: Performance gains: faster training or true improvement?**
> While this is a good question, completely decoupling these two performance-influencing factors is challenging. We attempt to provide some indirect insights into this issue. There are two perspectives: (1) additional visualization experiments demonstrating significant changes in transformer attention patterns (Appendix A.2 Figure 19), and (2) scaling curves for ResFormer and the vanilla transformer regarding model and data size (Figure 6). This common method [3] [4] verifies consistent and stable performance improvements in ResFormer. Notably, SVFormer's final performance is a little lower than the original Transformer due to its reduced parameter count.
>
> ### **Q3: Experiments on image recognition based on ViT.**
> Please refer to our response to **Reviewer dQDC ``Experimental Setup and Comparisons”**.
>
> ### **Q4: Reasons for dataset and architecture.**
> Llama is one of the best open-source models available and is widely used for study [5]. We have extended our SVFormer-Transformer comparison to other architecture like GPT-2 architecture, see Figure 15 (d). Besides, SlimPajama is the open-source corpus that most closely resembles Llama's pretraining data distribution and is really comprehensive. Notably, pretraining corpora from different domains do not significantly affect language model scaling trends [3].
>
>
> [1] Xiangming Gu, et al. "When Attention Sink Emerges in Language Models: An Empirical View." Arxiv(2024).
>
> [2] Mingjie Sun, et al. "Massive activations in large language models." arXiv (2024).
>
> [3] Jared Kaplan, et al. "Scaling Laws for Neural Language Models." Arxiv(2020).
>
> [4] Albert Gu, et al. "Mamba: Linear-time sequence modeling with selective state spaces." COLM(2024).
>
> [5] Haoyi Wu, et al. "Layer-Condensed KV Cache for Efficient Inference of Large Language Models." ACL(2024).

---

### Official Review · Reviewer_tUj8 · 2024-11-02

**Soundness:** 2
**Presentation:** 2
**Contribution:** 2
**Rating:** 3
**Confidence:** 2

**Summary:**

This manuscript presents a novel framework for approximating cross-layer attention. Within this framework, the authors introduce ResFormer as a practical implementation, demonstrating its effectiveness in mitigating attention concentration challenges. In addition, they propose SVFormer within the same framework, which further enhances efficiency by reducing the memory requirements for KV caching, thus lowering overall computational costs.

**Strengths:**

The manuscript proposes a framework for reducing the computational cost of cross-layer attention, offering a unified approach that integrates and extends existing methods, including NeuTRENO and DenseFormer.

**Weaknesses:**

1. The paper lacks discussion of prior work on the attention concentration problem and the connection to the over-smoothing issue addressed by NeuTRENO is unclear. A more detailed review of relevant literature would enhance clarity and better contextualize the impact of this work.
2. Using training loss as a criterion for comparing model performance is unconvincing (e.g. in Section 4.2, 4.3, 4.6), as it may not accurately reflect generalization. A more reliable evaluation metric, such as accuracy or perplexity on a separate validation set, would provide a clearer assessment of the model's effectiveness.
3. Minor comments:
- The term “gold attention matrix” in Section 4.3 should be clearly defined for better understanding.
- Right margin violated at line 659.
- Some references list only the first author; please ensure consistency in citation formatting.

**Questions:**

Could you provide a detailed comparison of training time and memory requirements for SVFormer and ResFormer relative to other baseline models? Such a comparison is crucial for understanding the extent to which these models benefit from the proposed framework.

---

> ### Author Response · Authors · 2024-11-24
> **Response to Reviewer tUj8**
>
> Thank you for your thoughtful review and suggestions. We address the concerns below:
>
> ### **Q1: Problem definition of attention concentration and relevant work.**
> Please refer to our response to **Reviewer dQDC ``Q1 Problem Definition and Motivation"** and **``Q2 Novelty and Theoretical Basis"**. Generally speaking, over-smoothing is an inherent model characteristic, while attention concentration emerges during optimization. These two phenomena are interrelated. Previous studies have well studied attention sink, a specific manifestation of attention concentration. Attention sink describes static features at a particular layer, whereas attention concentration represents dynamic changes in attention patterns as the model deepens. In practice, attention concentration and dispersion interweave as model depth increases. Existing approaches mitigate attention concentration by redirecting it from sequence tokens to additionally introduced bias terms, merely shifting its focus. In contrast, ResFormer and SVFormer alleviate attention concentration by preventing value-state drains, based on the principles and characteristics of how attention concentration occurs. NeuTRENO, however, fails to effectively influence attention pattern learning due to its use of first-layer values post-attention. Our new visualizations results support these findings, see Figure 1, Figure 2, Appendix A.1 Figure 18, Appendix A.2 Figure 19, Figure 20 and Figure 21.
>
> ### **Q2: Evaluation criterion is unconvincing.**
> Please refer to our response to Reviewer dQDC ``Experimental Setup and Comparisons". We evaluate our model using perplexity (loss) and assess its performance on several widely used downstream tasks. This approach aligns with common practices in language model evaluation. In Appendix A.6 Table 5 (Table below), we provide a detailed report of the loss for models of different sizes on various domain-specific validation sets. Additionally, the scaling curves in Figure 6 are also based on validation losses.
>
> | Model             | Common Crawl | C4    | Github | Stack Exchange | Wikipedia | Book  | Arxiv | Avg.   |
> |-------------------|--------------|-------|--------|----------------|-----------|-------|-------|--------|
> | Transformer (82M) | 3.3595       | 3.5388| 1.4247 | 2.3872         | 2.9047    | 3.3797| 2.1779| 2.7389 |
> | Transformer (180M)| 3.0961       | 3.2834| 1.2451 | 2.1651         | 2.5897    | 3.1309| 2.0001| 2.5015 |
> | Transformer (468M)| 2.8514       | 3.0430| 1.0908 | 1.9628         | 2.2821    | 2.8979| 1.8362| 2.2806 |
> | ResFormer (82M)   | 3.3362       | 3.5191| 1.3941 | 2.3592         | 2.8646    | 3.3572| 2.1518| 2.7117 |
> | ResFormer (180M)  | 3.0631       | 3.2504| 1.2200 | 2.1350         | 2.5435    | 3.0994| 1.9732| 2.4692 |
> | ResFormer (468M)  | 2.8214       | 3.0115| 1.0730 | 1.9388         | 2.2477    | 2.8696| 1.8142| 2.2537 |
>
> ### **Q3: Training time and memory comparison.**
> ResFormer does not increase memory usage compared to Transformer, either during training or inference. It requires retaining the value states from the first layer throughout forward propagation. During training, Transformer already stores these as intermediate activations, while during inference, these value states can be directly retrieved from the KV-cache. In terms of computational complexity, ResFormer introduces an additional matrix addition operation per layer. However, this extra computation is negligible compared to the matrix multiplications performed in each layer.
>
> We define $n_{layer}$ as the number of layers, $d_{model}$ as the dimension size of hidden state, $d_{ffn}$ as the dimension size of intermediate feed-forward layer and $l$ as the sequence length. Taking the GPT-2 model as an example, where $d_{ffn} = 4d_{model}$, the number of non-embedding parameters is $12 n_{layer}d_{model}^2$ [1]. SVFormer reduces the total parameter count by $(n_{layer}-1)d_{model}^2$, which is $\frac{n_{layer}-1}{12n_{layer}}$ of the original non-embedding parameters.
>
> ### **Q4: Other questions.**
> We have revised the references as your suggestions. Besides, the term 'gold attention matrix' actually means that we adopt cross layer attention and use $Attn(Q_i, K_1)$ rather than $Attn(Q_i, K_i)$ as mapping matrix for $V_1$. We recognize that this may have been misleading and have revised the relevant text and figure captions.
>
> [1] Jared Kaplan, et al. "Scaling Laws for Neural Language Models." Arxiv(2020).

---

> > ### Comment · Reviewer_tUj8 · 2024-11-26
> > **Response to Rebuttal**
> >
> > I would like to thank the authors for the rebuttal.
> >
> > A significant concern remains regarding the paper's reliance on training loss for comparing model performance and tuning hyperparameters, which I find unconvincing as a primary evaluation metric. While I appreciate the effort to address this by including validation loss in Table 5 and in scaling curves in Figures 6 and 15d, much of the presented results (e.g., Figures 5, 7, 8, 9, 10, 11, 13, 14, 15abc, 16, 17, and 22) are still based on training loss. I believe the paper would benefit from revising this issue; therefore, I maintain my score at reject.

---

### Official Review · Reviewer_2rAe · 2024-11-04

**Soundness:** 2
**Presentation:** 2
**Contribution:** 2
**Rating:** 5
**Confidence:** 4

**Summary:**

Paper proposes SVFormer, a way to reduce the size of the KV cache in Transformers by almost 50%. The authors propose sharing the values from the first self-attention layer across all layers. They find that this outperforms other approaches that reduce the KV cache size and perform extensive ablations to find when SVFormer works.

**Strengths:**

- Paper is straightforward and easy to read.
- It's interesting that values from the first layer can be used throughout the network for a small loss penalty.
- Authors thoroughly discusses prior work and explains the contributions of this work.
- Lots of ablations and experiments.

**Weaknesses:**

- The paper leaves out many important details. See the "Questions" section for specifics.
- Results are not well organized, and appear to have contradictory findings. Fig. 13 (c) in particular shows that SVFormer only outperforms a vanilla transformer when they have 2M parameters, which is very small.  At 82M parameters, SVFormer already is worse than the baseline. Fig. 13 (d), 14, and 15 also indicate that SVFormer hurts loss. However, Fig. 6 shows that SVFormer does better at larger scales
- I don't like the practice of subtracting the transformer performance and showing the difference. It potentially (a) hides bad baseline performance, and (b) potentially hides the fact that the difference between methods is tiny compared to the overall training loss curve.

**Questions:**

- Fig 4:
  - What model is this?
  - It seems very shallow -- only 6 layers?
  - These seem like such shallow models. Is “current mapping” Eq. 4 or Eq. 5?
- Eq. 8: why is the identity matrix  $J$ and not $I$?
- Effect of scale unclear:
  - Which figures correspond to the 700M parameter model described in 4.1.1?
  - How are the hyperparameters tuned for the baselines (especially the vanilla Transformer)?
  - Why is Fig. 6 (right) inconsistent with Fig. 13 (c) on the effect of model size?
  - Authors should show scaling laws to show much better or worse their method is they scale up their model.
- L460: "SVFormer will always be gradually surpassed by vanilla attention during training while its training speed is much faster than vanilla attention." How much faster can it be during training?

---

> ### Author Response · Authors · 2024-11-24
> **Response to Reviewer 2rAe**
>
> Thank you for your detailed review! Thank you for your interest in SVFormer. However, I believe you may have overlooked ResFormer. I look forward to hearing your suggestions of it.
>
> ### **Q1: Effect of scale is unclear.**
> We reorganized our results and presented the scaling trends of valid loss for both ResFormer and Transformer as model size and data size increase in Figure 6. Additionally, we illustrated the scaling behavior of SVFormer across different model sizes in Figure 15(d). The results demonstrate that ResFormer achieves equivalent validation loss with 10.4% fewer model parameters and 13.6% less training data compared to Transformer, while maintaining similar memory usage and computational cost. SVFormer, while reducing the KV-cache by nearly half, requires a 12.2% increase in parameters to achieve the same validation loss as Transformer.
>
> We have provided detailed hyperparameters for models of various sizes and different training sequence lengths in Appendix A.5, Tables 3 and 4. All our trained models have fewer than 1 billion parameters (approximately 700 million non-embedding). Results in Appendix A.4 show that with 20 billion training tokens, none of the models overfit. Note that we trained all models (NeuTRENO, DenseFormer, vanilla Transformer, and Resformer) under the same experimental conditions and then calculated the relative training loss. The parameters of SVFormer are less than Transformer. In Appendix A.3 Figure 22, we present the results of NeuTRENO under various values of λ. The optimal performance was achieved with λ = 0.4, which is just the model we used for comparison.
>
> ### **Q2: Relative training loss is not straightforward.**
> We utilized relative training loss to better illustrate the consistent benefits of new models throughout the training process. While absolute improvements are influenced by various factors such as learning rate, it is more important to demonstrate that the new models consistently provide stable benefits throughout the training process. We believe that the use of relative training loss more clearly highlights this trend of improvement. Additionally, we wish that the scaling experiments mentioned above can address your concern about the perceived small differences between methods.
>
> ### **Q3: Other questions.**
> 1. The results in Figure 6 and Figure 15(d) are not contradictory; they demonstrate the relative performance of ResFormer and SVFormer, respectively, against the standard Transformer.
> 2. We updated the Figure 4 to clarify: The lines illustrate the similarity of layer outputs using current attention (Eq. 4) or identity attention (Eq. 6) compared to those using cross-layer attention (Eq. 2). Using a trained 82M Transformer model, we investigate the output space differences when mapping values from the previous layer to the current layer's attention output using different matrices.
> 3. In Equation 8, I should be used instead of J to represent the identity matrix. Thank you for pointing this out.
> 4. SVFormer initially trains faster but achieves slightly lower final performance than the standard Transformer due to its reduced parameter count. We apologize for any previous imprecise statements.

---

### Author Response · Authors · 2024-11-24
**Updated Manuscript and General Response**

We sincerely appreciate all the reviewers for their time and constructive comments.

In general, we proposed **ResFormer** and **SVFormer**. Both of them mitigate attention concentration through **elegant and concise approaches**, whereas previous methods attempted to divert attention concentration by introducing additional tokens or similar techniques. Specifically, they alleviate value state drains by incorporating the first layer's value into the current layer's value, thereby indirectly suppressing attention concentration. ResFormer achieves equivalent validation loss with **10.4%** fewer model parameters and **13.6%** less training data compared to Transformer, while maintaining **similar memory usage and computational cost**. Besides, SVFormer, while reducing the KV -cache by nearly half, requires a 12.2% increase in parameters to achieve the same validation loss as Transformer.

In light of the reviewers' feedback, we have made several enhancements to our manuscript. **All changes are highlighted in blue in the updated PDF**. A summary of the **key updates** is provided below:
1. **Figure 1**: Add illustration of the over-smoothing and attention concentration in randomly initialized models; add illustration of the attention concentration issue in Llama and Mistral models; and substitute the entropy analysis of 82M models with that of 468M models.
2. **Figure2**: Add the analysis of token importance, value-state norms, and hidden-state norms of the first token across layers of 468M models (Transformer, NeuTRENO, ResFormer, SVFormer).
3. **Section 1**: The connections between over-smoothing, attention sink, and attention concentration are elaborated in detail. Recent research progress related to attention concentration is introduced. The performance and novelty of the model are emphasized.
4. **Figure 6 & Figure 15**: Validation loss scaling curves for ResFormer (SVFormer) and Transformer are shown, comparing their performance across model and dataset sizes.
5. **Figure 8 & Figure 22**: Robustness analyses of hyperparameters are conducted for both ResFormer and NeuTRENO.
6. **Section 4**: Add analysis of scaling results and visualization outcomes.
7. **Appendix A.1**: Visualization of the over-smoothing phenomenon is added.
8. **Appendix A.2**: More visualization of the attention concentration phenomenon is added.
9. **Appendix A.5**: Detailed hyperparameters for models of various sizes and different training sequence lengths are added.
10. **Appendix A.6**: Validation loss results for various models is added.

---

### Meta-Review · Area_Chair_5fQH · 2024-12-20

**Metareview:**

All reviewers converged on rejecting the paper post rebuttal. The AC checks all the materials, and while appreciating the additional efforts including results and analyses and making major modifications to the draft, the AC resonates with the reviewer consensus that the paper currently has issues to address and would benefit from another cycle.

**Additional Comments On Reviewer Discussion:**

Reviewers have remaining concerns on:
- Extensive reliance on training loss (instead of validation loss) is not convincing -- to which the AC *disagrees* as all the models are trained on the SlimPajama dataset for *one* epoch only, which means all the losses are computed on the *unseen* data even if it is on training. However, the authors failed to point this out.
- Motivation of using cross-layer attention is less clear, especially given cross-layer attention underperforms compared to ResFormer. Reviewers are concerned about the potential conflicting signal.
- Lack of more rigorous experiments on decoupling faster convergence and actual accuracy improvements. Reviewers are not satisfied with the answer.

---

### Decision · Program_Chairs · 2025-01-22

Reject